# Is Voting for a Cartel a Sign of Cooperativeness?

Joris Gillet 

Department of Economics, Business School, Middlesex University, The Burroughs, London NW4 4BT, UK; j.gillet@mdx.ac.uk

**Abstract:** This paper tests the hypothesis that a (partial) reason why cartels—collective but costly and non-binding price agreements—lead to higher prices in a Bertrand oligopoly could be because of a selection effect: decision-makers who are willing to form price agreements are more likely to be less competitive and pick higher prices in general. To test this hypothesis we run an experiment where participants play two consecutive Bertrand pricing games: first a standard version without the opportunity to form agreements; followed by a version where participants can vote whether to have a (costly) non-binding agreement as a group to pick the highest number. We find no statistically significant difference between the numbers picked in the first game by participants who vote for and against an agreement in the second game. We do confirm that having a non-binding agreement to cooperate leads to higher numbers being picked on average. Both participants who voted for and against the agreement increase the number they pick in situations with an agreement. However, this effect is bigger for participants who voted in favour.

**Keywords:** social dilemma; oligopoly; non-binding promise; experimental economics

**JEL Classification:** C91; D02; D43; L13

## 1. Introduction

Collective but non-binding agreements to act cooperatively have been shown to be effective and lead to better group outcomes in numerous economic situations. From a theoretical perspective such agreements should have no effect on subsequent decisions. In most settings they are unenforceable and costless to break and do not affect the pay-off structure of the game. They are, as such, nothing more than cheap talk because they do not change the dominant strategy. Still, these kinds of collective agreements to act in the best interest of the group have led to higher prices in oligopolies (for instance Apesteguia et al. [1], Gillet et al. [2]), less extraction in the common pool dilemma (Mosler [3]) and increased cooperation in different social dilemmas (Hopthrow and Abrams [4], Hopthrow and Hulbert [5]). Papers like Dannenberg [6] and Kroll et al. [7] show the limits of the effect of non-binding collective agreements by pointing out that in public good games these agreements are only really effective in the first few iterations. This paper tests the hypothesis that a (partial) reason why collective agreements are effective, may be because of a selection effect: that the type of players who are willing to form these agreements are more likely in general to perform the behaviour that is being promised.

Collective agreements within a group can be seen as a special case of promise making. The literature that investigates why promises work focuses on two mechanisms. First of all, the positive effect of promises seems driven by a form of guilt aversion. If someone is interacting with someone who is expecting a particular behaviour from them—if they have been made a promise for instance—this first person may experience negative utility from disappointing the second person which makes them more likely to follow up on the promise made (Charness and Dufwenberg [8], Ederer and Stremitzer [9]). The second factor appears to be a personal preference for consistency between, in this case, statements about intentions and actions. In some very smart experiments Vanberg [10] and Ismayilov

and Potters [11] find that promises are kept even when they are not received by the person who is affected by the decision the promise is made about. Ellingsen and Johannesson [12] also provide experimental and theoretical evidence for the role of consistency preferences.

A third, additional factor might be that people who are more likely to perform a particular behaviour are more willing to make a promise to perform that behaviour in a future interaction. Koessler et al. [13] and Koessler et al. [14] find evidence for this selection effect for promise makers. People who are willing to promise to pay their taxes on time are more likely to have paid their taxes on time previously (Koessler et al. [13]) and people who are willing to promise to cooperate in a public good game were more cooperative in a previous public good game where there was no possibility to make promises (Koessler et al. [14]). The selection effect cannot explain all of the positive effect of the promise (in Koessler et al. [14] for instance, compulsory promises are also effective) but suggests that a portion of the total effect is because the promise to cooperate is being made by people who are more likely to cooperate in general.

Dannenberg and Martinsson [15] also find evidence for this selection effect for promise makers in the context of a public good game. They find that people who are willing to go into a collective agreement to cooperate are indeed more cooperative in general. They find a positive difference for contributions in a previously played regular public good game without the opportunity to form any agreement. They also find a positive difference in the situations in the game with the opportunity to form an agreement but where the group decided not to go into an agreement. However, the overall effect of the non-binding agreement in Dannenberg and Martinsson [15] is fairly small. One possible reason for this is that they use groups of people who already know each other outside of the context of the experiment. The baseline cooperation level, in the version of the game without any agreement opportunity, is already remarkably higher than in other studies.

One difference between Koessler et al. [14] and Dannenberg and Martinsson [15] is that the first looks at individual promises—participants individually giving public (but non-binding) statements pledging to perform a particular future behaviour—and the latter look at collective, group-level (but still non-binding) agreements to behave a certain way in the future. In many situations, promises are of such a collective nature (Orbell et al. [16]). International treaties like the Paris Agreement for instance, are group level agreements to cooperate (that are, because of their international nature, not readily enforceable) (Barrett [17]).

An archetypal and important real-world example of such a collective agreement is a cartel in an oligopoly market. A cartel is a situation where the different players in a market come together and agree on a communal course of action that is beneficial for all of them (limiting supply, increasing prices). Since cartels are usually illegal, the cartel agreement is unenforceable. You cannot easily go to court to sue one of the other cartel members if they did not adhere to the agreement. (Cartels are also possibly costly because of the chance of having to pay a fine when they are discovered). The positive effect of cartels for oligopoly firms has been shown experimentally in both Cournot (Waichman et al. [18], Fonseca and Normann [19]) and Bertrand games (Hinloopen and Soetevent [20], Chowdhury and Crede [21], Chowdhury and Wandschneider [22] in addition to Apesteguia et al. [1] and Gillet et al. [2]).

Hinloopen and Soetevent [20] in fact hypothesize the existence of the selection effect, like the one Koessler et al. [14] and Dannenberg and Martinsson [15] observed in the public good game, in the Bertrand game. They observe that, in Bertrand experiments with the opportunity to form cartels, competition is higher (and prices picked lower) in situations where a group of firms decide not to form a cartel compared with a benchmark situation where there was not the possibility to form a cartel to begin with. They suggest that this may be caused by a selection effect; that a group of participants who are on average more competitive are less likely to be willing to form a price agreement.

The goal of this paper is to test the selection effect hypothesis in the context of a Bertrand pricing game (Dufwenberg and Gneezy [23]). The Bertrand pricing game is a

simple model of oligopoly (price) competition in a market with homogenous goods. The game is an example of a social dilemma. The socially optimal outcome is where all players pick the highest possible price and share the maximum total earnings, but every individual player has an incentive to undercut their competitors and pick a lower price. The (one shot) Nash equilibrium is the situation where everyone picks the lowest number. The specific hypothesis this paper tests in a Bertrand pricing game is that (firstly that cartels work, and lead to higher overall prices, and secondly that) the players who are willing to form cartels—and are willing to make the collective promise to pick the highest price—pick higher prices in general.

Even they though are both social dilemmas, the strategic cooperative considerations in the public good game—as used by Koessler et al. [14] and Dannenberg and Martinsson [15]—and the Bertrand pricing game are somewhat different. Cooperation in the public good game is about social preferences. By contributing a bit more and foregoing some of their own earnings, a player can increase the earnings of the others in the group. Additionally, this process can be, depending on the design of the experiment, usually be fine-tuned to fit the social, cooperative preferences of the decision maker. The more you contribute in a public good game, the more cooperative you are. Picking a higher number in the Bertrand pricing game is not necessarily a sign of cooperativeness. It is only so if it is part of a (either tacit or explicit) strategy to pick the same high number as the other firms in the market. Picking a high number can be uncooperative if the decision maker expects this number to be just below the prices picked by the other firms. As such, the interpretation of the number picked in the Bertrand pricing game is more complex than that of the contribution in the public good game. Its meaning depends on the associated expectations, but it is also the case that the lower the number picked, the more competitive the action.

The question being investigated here is, is the selection effect for promise makers—shown by Koessler et al. [14] and Dannenberg and Martinsson [15] to be present for contributions in the public game—also a feature for competitiveness in the Bertrand pricing Game? Are players who are willing for form cartel—to form a collective agreement to act less competitively—generally less competitive, as expressed by the higher prices picked? To test this hypothesis, we run an experiment where participants play two Bertrand oligopoly games in a row. First a standard version, without the opportunity to form a collective agreement. This is followed by a version where participants can vote, by majority, on whether to have a (costly) non-binding collective agreement to pick the highest number. We find no significant evidence that participants who vote for an agreement in the second game picked higher numbers in the first game. We do confirm the finding that having a price agreement leads to higher numbers being picked and also find that participants who voted for or against the price agreement behave differently in response to the existence of the price-agreement. Both For- and Against-voters pick higher numbers in situations with a price agreement but participants who voted for a price-agreement react more strongly to the existence of the price agreement.

## 2. Methods

The Bertrand pricing game used in the experiment worked as follows: there are three participants who each, simultaneously, pick a number between 1 and 50. The winner of the game is the participant who picks the lowest number and their earnings, in experimental points, are equal to the number that they picked. The other participants earn nothing. If two or more participants pick the same lowest number, the earnings are shared equally.

Participants played two consecutive Bertrand pricing games. The first game was the basic game as described above. After picking their number for the game, participants were also asked their expectation about the numbers being picked by the other participants in their game by guessing the average number being picked by the other two players. This question was incentivized by giving a correct—or almost correct, a difference of plus or

minus 5 was close enough—answer a bonus payment. Participants did not receive feedback on the outcome of the first game at this point in the experiment.

After the first game participants were told they were going to play one more, final, game, with a different set of players than the first game. The rules of the game were going to be the same but now, before making their decision what number to pick, participants would have the opportunity to make a collective agreement with their co-players to pick the highest number. The group decision was by majority. When 2 or more of the group members voted for the agreement, the agreement was to be in effect. The agreement was always to pick the highest number but was non-binding; group members could still pick any number they wanted. There was a probabilistic cost associated with the agreement. If the group formed an agreement there was a 20% chance that the results of the subsequent decision would not count. In that case, nobody in the group would earn any points. The idea behind this feature is that it mimics cartels being illegal (and potentially leading to fines if discovered) in the real world. Additionally, associating a (small) cost with the making of the promise made sure that not everyone made a promise. For the statistical analysis we would like to have an as even as possible distribution of For- and Against-voters. Participants in the second game made three decisions: whether they voted for or against the agreement and, applying the strategy method, a price for a situation where the group decided for an agreement and a price for a situation without an agreement. The instructions (available in the Appendix A) were framed in a neutral and abstract manner. There was no mention of 'prices' or 'cartels'.

The experiment was run on Amazon Mechanical Turk. As running interactive group experiments online is rather complicated (Arechar et al. [24])—coordinating all participants to be, and stay, online at the same time, etc.—we used post hoc group formation. Participants made their decisions separately and in their own time. Only afterwards, once all participants had made their decisions and the data was collected, did we combine the participants into groups and played out their interactions, and calculate their earnings, according to the decisions. Only after this process had happened, we could tell, and pay, participants their earnings. The experiment was approved by the ethical review board of the Middlesex University business school.

After reading the instructions of the first game, participants had to answer two comprehension questions to show that they had understood the game. Participants received a participation fee of USD 1.00 and each point they earned was worth an additional USD 0.03. The bonus payment for guessing the average number picked by the other members of their group correctly was USD 0.25.

With respect to the appropriate number of participants we take the results of Kiessler et al. [14] as a guidance to establish the effect size. The difference in cooperativeness in their first game between promise-makers and non-promise-makers in the second game suggest a Cohen's D of 0.65. To obtain power of 0.90 with $\alpha = 0.05$ to detect an effect of $d = 0.65$, the required sample size is at least 51 observations per treatment for a two-tailed analysis

## 3. Results

In total 378 participants took part in two sessions, in August 2015 and March 2016. Average earnings, including the participation fee, were USD 1.55. In the following analysis we will only look at the decisions made by the 362 participants who answered both comprehension questions correctly. Of these 362 participants 152 were female and 209 were male (one participant did not answer this question). The average age was 33.99.

Of these 362 participants, 183 voted for a price agreement and 179 voted against. When we look at the numbers they picked in the first, basic Bertrand game we see that the For-voters picked on average 21.69 (sd = 11.07) and the Against-voters 19.67 (sd = 10.62). The difference is not statistically significant ($p = 0.1416$). (All statistical comparisons between two averages in this paper are based on two-tailed Mann–Whitney tests).

Figure 1 shows the cumulative distribution of numbers picked in the first game by the two types of voters. A Kolmogorov–Smirnov test cannot reject the hypothesis that the two distributions are the same ($p = 0.57$).

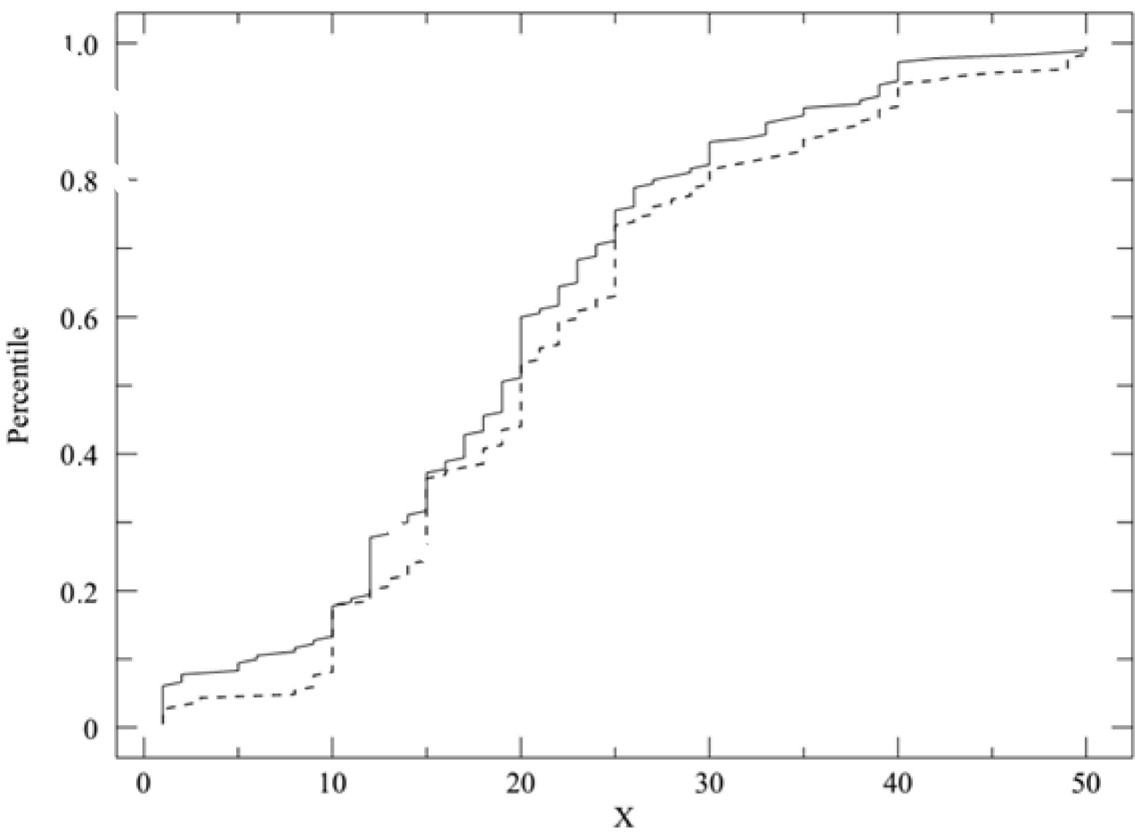

**Figure 1.** Distribution of numbers picked by For-voters (continuous line) and Against-voters (dashed).

Expectations about what number was picked, on average, by the other members of the group were higher than the number the participants picked themselves: 25.62 (sd = 10.75) vs. 20.70 (10.88). This difference is statistically significant ($p < 0.0001$). There is a high correlation between expectations and actual number picked ($R^2 = 0.617$, $p < 0.0001$). Therefore, when we look at the difference in expectations between For- and Against-voters we see very similar results as for the actual choices: average expectations are 26.61 (sd = 11.95) vs. 24.60 (sd = 10.41), respectively. This difference is not significant ($p = 0.12$).

Another potential way to interpret the hypothesis is to look at the relationship between number picked and expectations for For- and Against-voters separately. As said, picking a higher number is only really cooperative if it is part of a collusive strategy, meaning when the player picks the same number as they expect the other players to pick. A high number can also be competitive, as long as it is slightly below what the player expects other players to do. This suggests that for cooperative players the difference between number picked and expectation should be zero and for competitive players larger (but still small). When we compare these differences for For-voters (4.92, sd = 7.62) and Against-voters (4.92, sd = 6.52) we see that they are almost identical (and both significantly larger than zero, $p < 0.0001$).

If we assume highly competitive players to be relatively more strategic decision-makers, they would maximize their profits by picking a number slightly below their expected number. Meaning we would find a high correlation between expectation and number picked. If we think of not-so-competitive players as being less strategic we would maybe expect a lower correlation between expectation and number picked. The correlation

between expectation and number picked is indeed lower for For-voters ($R^2 = 0.76$) than for Against-voters ($R^2 = 0.81$), but this difference is not significant ($p = 0.11$).

There is also no difference between For- and Against-voters with regard to age. The average age of For-voters was 33.83 (sd = 9.18) and of Against-voters 34.16 (sd = 9.58) ($p = 0.93$). Nor is there a significant effect of gender. Women voted for and against 70 to 82 and men voted 112 to 97 ($p = 0.16$ in a Chi-square test).

Even though it is non-binding the agreement leads to significantly higher numbers being picked. In the second game, when asked to pick a number for situations with an agreement, participants pick on average 40.91 (sd = 12.92). For situations without an agreement they pick on average 22.97 (sd = 11.42). This difference is significant, $p < 0.001$.

Both For- and Against-voters pick a higher number in situations with an agreement, but the size of this effect is different. For-voters react more strongly to the existence of the agreement. For situations without an agreement there is no significant difference between For- and Against-voters: 23.59 (sd = 11.372) vs. 22.34 (sd = 11.464), $p = 0.35$. However, for situations with an agreement For-voters pick on average a significantly higher number than Against-voters: 45.25 (sd = 9.051) vs. 36.47 (sd = 14.693), $p < 0.001$. Figure 2 summarizes these findings. The difference between number picked with and without an agreement is 21.66 (sd = 13.32) for For-voters and 14.13 (sd = 15.38) for Against-voters. This difference is statistically significant ($p < 0.0001$).

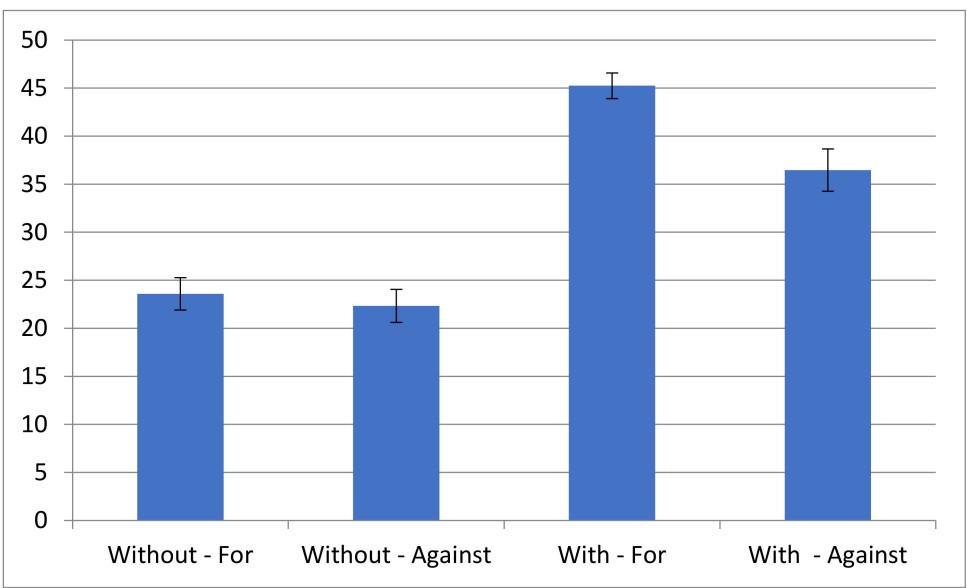

**Figure 2.** Average number picked in situations with or without an agreement for participants who voted for or against the agreement (with 95% confidence bars).

Even though the agreement leads to higher numbers being picked on average, this does not mean that all players adhere to the agreement they have made and keep their promise to pick the highest number. In total 47% of the participants (170 out of 362) did indeed pick the highest number in the second stage in the case of an agreement. This fraction is, not surprisingly, higher for people who voted for the agreement (116 out of 183) than for those who voted against (54 out of 179).

We can also use this data to test whether adhering to the agreement is related to lower general levels of competitiveness. Those participants who voted in favour of the agreement and subsequently kept it, picked on average 22.18 (sd = 11.16) in the first game. Those who voted in favour of the agreement but subsequently broke it by picking a number lower than the highest possible, picked on average 20.85 (sd = 10.94) in the first game. This difference is not statistically significant ($p = 0.2$). For Against-voters the difference is significant ($p = 0.047$)—promise-keepers pick on average 21.94 (sd = 11.2) and -breakers

18.7 (sd = 10.26)—but the interpretation of this result is somewhat convoluted: players who do not want to form an agreement to collude but are forced to do so anyway by the group majority and despite this still adhere to the agreement, are more cooperative than those who break their (forced) promise. One way of reading this result might be that people who are more likely to go with the will of the group (against their own preferences) are more cooperative. (As one of the referees has pointed out, comparisons between For- and Against-voters in different situations the second game need to be considered with some level of caution. Combining the players own vote and the outcome of the voting procedure, will lead to differences in information about the intentions of the other players in the group and, presumably, different expectations. Additionally, because in the current experiment the cartel is decided by majority, its effect cannot be readily compared to that in other, previous studies, where cartels are usually decided unanimously).

## 4. Discussion

The experiment described in this paper fails to find evidence for the hypothesis that a (partial) reason why (costly) collective non-binding agreements to be less competitive lead to higher prices being picked in a Bertrand pricing game, is because these agreements are made by people who are less competitive (and pick higher prices) in general. Participants who voted for the price agreement in the second game do not pick significantly higher numbers in the first game than people who voted against.

This result goes against the hypothesis of Hinloopen and Soetevent [20] and the findings of Kiessler et al. [14] and Dannenberg and Martinsson [15]. These latter two papers find that people who are willing to make a promise to cooperate in a public good game, are more likely to have to contributed more in a previous public good game where there was no opportunity to make a promise. The current findings suggest that the selection effect that they propose—a partial reason why promises work is because they are made by people who are already more likely to perform the behaviour that is being promised—is not a universal feature for all types of promises and all types of behaviour. It does seem to play a role in cooperative behaviour in the public good game, but not in (non)competitive behaviour in the Bertrand pricing game.

This difference with the results from the public good game can probably be attributed to the fact that the act of making a promise plays different roles in the public good game and the Bertrand pricing game. In the public good game, a willingness to agree to contribute more can be part of the expression of a cooperative personality. In the Bertrand pricing game this link between willingness to form an agreement and general behaviour is less straight forward. For a cooperative person it is difficult to express their cooperativeness in the regular version of the game (picking a higher number is not necessarily cooperative) and even strategic self-interested profit maximisers might have a reason to vote for a price agreement (it might increase their profits if other players adhere to it and they manage to undercut them).

Another piece of evidence for this interpretation is that the results of the current experiment show that the collective agreement also has an increasing effect on the number being picked by group-members who did not vote for it. This is also in contrast to Kiessler et al. [14] where the participants who did not make the promise show the same level of (low) cooperation with the agreement as without. This is consistent with the idea that in the public good game For-voters are more cooperative in general (or, maybe more accurately, that Against-voters are less cooperative in general). In Dannenberg and Martinsson [15] the agreements are formed by unanimity so there are no situations where Against-voters are in a group with an agreement.

The finding in the current experiment that Against-voters react positively to the existence of an agreement—pick a higher number than in situations without, even if they did not vote for the agreement—does not necessarily mean that they are acting cooperatively. The difference could be the result of strategic, competitive profit-maximizing considerations. As agreements are formed by majority, an Against-voter knows that, in

a situation where the group enters into an agreement, the two other group members will have voted in favour. It is not unreasonable to assume that this might influence the Against-voters to increase their expectations about what number the two other group-members will pick. Since the profit-maximizing strategy is to try and pick a number slightly lower than the other group members his expectation creates an incentive—to win but also to earn more when winning—to pick a higher number.

**Funding:** This research received no external funding.

**Institutional Review Board Statement:** The experiment was approved by the ethical review board of the Middlesex University business school.

**Informed Consent Statement:** Informed consent was obtained from all subjects involved in the study.

**Data Availability Statement:** The data from this experiment is available here: https://osf.io/8nj2v/ (accessed on 20 March 2021).

**Conflicts of Interest:** The author declares no conflict of interest.

## Appendix A. Experimental Instructions

Please read carefully. There will be a couple of questions to check if you understand them completely on the next page.

In the first part of this experiment you are part of a group. All the group members make their decisions separately from each other. Only after every participant has made their decision we can calculate what happens in a particular group. So you will learn the outcome of the group process (and your additional earnings) after the experiment has finished. You will never learn who the other people in your group were; just the consequences of their decisions. (And, vice versa, the other participants in your group will never know who you are).

To keep things simple we are playing for points in the experiment. Every point you earn during the experiment will be worth H\$0.03. In this first part you are part of a group of 3 players. Your task is to pick a (whole) number between 1 and 50. The winner is the player who picked the lowest number and their earnings are equal to the number they picked. If two or more players pick the same lowest number, the earnings are shared equally.

An example (with numbers picked for clarification purposes only). Let's say you pick 44. One other player in your group also chooses 44 and the second other player 50. The lowest number is 44. This is picked by you and one other player. You share the earnings and you each earn 22 points (=44 divided by two). The second other player, who picked 50, doesn't earn anything.

Another example. Let's say you pick 24, one other player picks 18 and the second other player 37. The lowest number is 18. That was picked by one of the other players. This player earns 18 points. You don't earn anything in this scenario. The second other player, who picked 37, also doesn't earn anything.

Before we start the actual experiment we would like to make sure that you understand the rules. Please answer the following two questions:

1. Imagine a situation where you pick 28, one other player also picks 28 and the second other player picks 36. How many points will you earn?

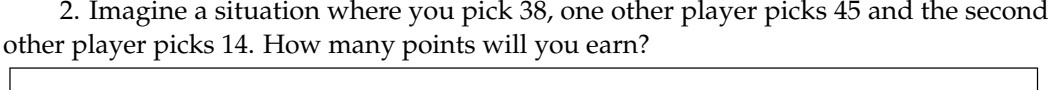

2. Imagine a situation where you pick 38, one other player picks 45 and the second other player picks 14. How many points will you earn?

Now we start with the real experiment. As a reminder: you are part of a group of 3 players. Your task is to pick a (whole) number between 1 and 50. The winner is the player who picked the lowest number and their earnings are equal to the number they picked. If

two or more players pick the same lowest number, the earnings are shared equally. Every point you earn during the experiment will be worth H$0.03.

I pick number:

```
[                                                                           ]
```

As said, you won't learn the choices by the other participants in your group until after the experiment is finished. However, we would like to know what you think the others will pick. What do you think the average number picked by the other two participants in your group will be? You'll get an extra bonus of H$0.25 if your guess is right or almost right (you can be off by 5 either way).

I expect the average number picked by the other participants in my group to be:

```
[                                                                           ]
```

In the second (and final) part of the experiment you will be part of a different group of three players. This will be entirely different people than the people in your group in the first part.

The game is the same. You'll pick a number between 1 and 50 and the winner is the player who picked the lowest number. Earnings will be equal to the number they picked and if two or more players pick the same lowest number, these earnings will be shared equally.

There is one extra thing: before you pick your number you will be asked if you want to make an agreement with the rest of your group to pick the highest number. There will only be an agreement if the majority of the players in your group (so 2 or 3 players) are in favour of it. The agreement is simply a collective statement that says that all members of the group promise to pick the highest number.

You don't have to adhere to the agreement. You can pick any number you want.

There is a potential risk associated with having an agreement. If a group decides to have an agreement there is a 20% (1 in 5; to be determined after your decisions) chance that the results of the subsequent decision don't count. If this is the case none of the group members will earn anything.

So, this second part consists of three decisions: a vote on whether you want to make an agreement in your group or not, the number you pick in the case your group makes an agreement and the number you pick in the case of no agreement. As in the first part, every group member makes their choices separately and we determine what happens in the group after everybody has made their decision.

I vote (for/against) an agreement to pick the highest number.

○   For
○   Against

Pick a number for situations where the group has decided to form an agreement to pick the highest number.

```
[                                                                           ]
```

Pick a number for situations where the group didn't decide to form an agreement to pick the highest number.

```
[                                                                           ]
```

Thank you for your cooperation!

Once we've collected the responses from all participants we can calculate the group outcomes and we'll be able to let you know (and pay you) your extra earnings.

We would appreciate it if you could also please provide the following demographic information.

Gender

○　Male
○　Female
○　Rather not say

Age (leave empty if you'd rather not say)

| |
|---|

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
