# Peer review of "Is Voting for a Cartel a Sign of Cooperativeness?"

_games, doi:10.3390/g12020048_

Round 1

Reviewer 1 Report

The paper studies a Bertrand competition game. There are two within-subjects conditions: a baseline game with no pre-play agreements and a game with pre-play non-binding agreements. Agreements take the form of a majority-voting procedure. If at least two out of the three group members vote for the agreement, they are informed that the group has an agreement to pick the highest number. Subjects play this second game in strategy-method, picking a number for both the situation where the group reached an agreement and for the situation where the group did not reach an agreement. The paper tests three main hypotheses (labels in parentheses are mine). First, whether subjects that vote in favor of the agreement pick higher numbers in the baseline game (H1). Second, whether the difference between the numbers they pick when an agreement is in place and those they pick in the baseline game is larger for subjects that vote in favor of the agreement  than for subjects that vote against it (H2). Third, within groups that voted similarly (in favor, H3a, or against, H3b), whether the same difference is larger for subjects that fulfilled the agreement than for those that broke the agreement. The results show a significant difference for H2 and H3b, but not for H1 and H3a. The Authors interpret their data as showing that the reason agreements typically work in social dilemmas does not go through selection of cooperative subjects. 

The experiment is competently implemented and described. The analysis is basic, but sound. The paper reads well. My main concerns regard the interpretation, which I think does not follow from the results. Overall, while the experiment is interesting, I do not think it can answer the declared research question.

My overall appraisal is that the paper is informative about what happens in a Bertrand game when we allow for these majority-voting agreements, and about the different behavior of people that vote in favor or against the agreement. I do not see how it could speak to the declared goal of testing "the hypothesis that a (partial) reason why non-binding agreements work is because the people who are willing to form those agreements are more cooperative in general".

  1. the main issue regards what does 'cooperative' mean in a Bertrand game. The problem lies in the different best response functions in such a game with respect to, e.g. a public good game. To make an example, choosing 20 when you believe others will also choose 20 is cooperative. Choosing 49 when you believe others will choose 50 is not cooperative. Therefore one cannot say that choosing higher numbers is 'more cooperative' than choosing lower ones, as the Authors seem to imply throughout the paper. The Authors are aware of this issue and mention it at the end of the discussion. However it is crucial for the interpretation of their results. It is particularly problematic when we compare choices across institutions that are likely to affect beliefs differently (and actually do so if the hypotheses are true). I do not think that the Authors can do much to address this concern. My suggestion is that they adapt the motivation and the interpretation so that they are aligned with what the data can actually say, abandoning aims that the design cannot meet.
  2. One thing the authors can do is to check whether voting behavior predicts the difference between actions and beliefs in the baseline game.  This is only a partial fix: the Authors collected beliefs about the average of others choices, we would like to have beliefs about the minimum between the partners choices, because that is what matters for the best response function.
  3. The information players have when they form an agreement is different depending on what they voted.  A subject that voted against knows both partners voted in favor of the agreement. A subject that voted in favor does not know if both partners voted for the agreement or not. This is a potential confound when comparing the behavior conditional on reaching an agreement, and it should be acknowledged and discussed. For instance it could easily imply that subjects that voted in favor have lower first-order beliefs than subjects that voted against. 
  4. I do not understand the role of adding risk to agreements (the fact that anyone may earn zero if they reach an agreement with some exogenous probability). It is not clear how it may interact with the selection of specific types. Will it favor selection of people who really want to cooperate? Will it selects people who are willing to take bets to make large profits? In any case it is another confound, and the text does little to explain the reason for this institutional arrangement.
  5. Deciding through majority voting and making a promise/declaring one's intended action are very different things. A discussion of these differences is missing. Why did the Authors choose such an institutional arrangement? 
  6. As I said above, my suggestion is to avoid talking about the issue of selection of cooperative types. If, however, the Authors do not agree with this advice, they should at least be more precise about how this selection would work in past experiments and how this maps in their own. My feeling is that it has to do more with signalling of one's type and coordination, rather than selection as typically (and narrowly) intended. After all, matching is always exogenous in these games.  

Minor:

p.3 typo: "make form a collective agreement"

p.4 "The experiment took not more than a couple of minutes to complete." Could the Authors be more precise? Taking 'couple' literally, I could not go through the instructions in a couple of minutes. 

The text says the action space is {0,..,50}. The instructions say it is {1,..., 50}. Please clarify.

Talking about cooperative types in the context of behavioral IO is always tricky. Many consider it hard to bridge the gap between experimental subjects and real-world firms when it comes to the utility function. The Authors should clarify if they consider their Bertrand game an abstract example of interesting game or if they intend their results to matter for oligopoly research. In the latter case, some support for this external-validity claim should be provided.      

Reviewer 2 Report

This paper reports an experiment investigating whether the effect of non-binding agreements on a price choice in a Bertrand competition increases the degree of “cooperativeness” because people making non-binding agreements are more cooperative to start with. The paper finds that this is not the case and hence does not replicate the results found in PG games by Dannenberg and Martinsson (2015). While I think the paper is interesting, I also have a number of concerns:

  • First of all, I have some doubts on the exact research question. Is this only a question of heterogeneity effects? I think the authors should motivate better the importance of the research question.
  • I wonder why the authors use the terms cooperativeness and cooperation; there is no externality here. I would rephrase this as collusion. Being a Bertrand competition it would be better to call it collusion to highlight the difference with the PG game in Dannenberg and Martinsson (2015). The authors note that their setup can be called coordination at some point in the paper. I think the latter is incorrect as coordination typically represents setups with multiple equilibria, while in the Bertrand game there is only one equilibrium.
  • One additional issue is that the authors refer to a literature on non-binding agreements in PG games but do not review the literature on communication and collusion in Cournot or Bertrand games. This would be important to evaluate the novelty of the current paper.
  • The procedures of the voting mechanism are not well explained in the paper. Only reading the instructions I realized that the agreement is related only to the highest number. Please explain better that this is the only possible agreement. That is, people cannot agree on a number different than 50.
  • What is the reason to introduce the feature that with 20% probability the decisions of the second interaction do not count? This is not explained in the paper. I am wondering about the effect of this mechanism on the willingness to vote in favour of an agreement. This mechanism renders the current paper not comparable with Dannenberg and Martinsson (2015) who do not have this option. Moreover, it has likely effects on the selection of participants who vote for the agreement. This selection could be based on risk aversion or other characteristics that are not controlled for.
  • Why is an agreement reached only if 2 out of 3 players agree? Can we really call this “agreement”? This is another element that renders the current paper not comparable with Dannenberg and Martinsson (2015).
  • The overall absence of evidence for the hypothesis of the authors let the reader speculate why is this the case. One immediate explanation as the authors suggest is that the Bertrand competition is different from the PG game. However, as the comparison with the PG game is confounded by the many differences with Dannenberg and Martinsson (2015) highlighted above, I think the authors should try to conduct the same experiment using a PG game instead of a Bertrand game (keeping constant all their modification of the original Dannenberg and Martinsson (2015) design). In that case, they could reliably conclude that it is the structure of the game that makes non-binding agreements more or less effective. As it is now, it is impossible to know whether it is the Bertrand game, the 20% probability of not being paid (point 4), or the different voting mechanism (point 5).

Round 2

Reviewer 1 Report

The new version improves substantially on the previous one. In particular I find it more straightforward on what we can and we cannot infer from the experiment. 

Overall I consider this a nice exercise and I am happy to recommend accepting the paper for publication. 

I found one typo in the first line of Section 2 Methods: "pame"-- > "game"

Author Response

Thanks again for your constructive feedback. It's made the paper much better. Thanks also for spotting that one remaining typing error.

Reviewer 2 Report

I think the paper is much improved. It explains properly the difference between Bertrand competition games and PG games and places the paper in the right context. However, I still have some concerns.

  • I am still uncertain about the exact mechanism the author hypothesizes. The author says that the effectiveness of the non-binding agreements to increase collusion/cooperation is through a selection effect which is distinct from the two reasons people have provided before (p. 2 “A third additional factor…”). Hence, their claim is that people who are likely to form the agreement are those who are generally more cooperative. This is fine but I think it cannot alone explain an average effect on overall collusion/cooperation in an agreement.

Maybe an example would make it clear. Let us assume three players are playing two PG game with the same structure of the author’s experiment. They have 20 tokens they can contribute to the public good. In the first game they contribute 0, 10, and 20. Positive contributions are explained by some model of social preferences (say for example warm glow). In the second game the participants who contributed 10 and 20 vote for the agreement and the other does not. Given the agreement is not binding and given their preferences, they will still contribute 0, 10 and 20 in the second game. Hence, selection in terms of social preferences or willingness to cooperate cannot cause any increase in contributions. I think some form of preferences for promise-keeping is necessary. Hence, the real reason for average higher contributions in the second game cannot be related to selection on social preferences, but maybe selection on social preferences plus promise-keeping. People who vote for an agreement are people who typically follow promises more than others. But this seems not to be the hypothesis of the current paper. I would like to see the exact mechanism the author has in mind better spelled out in the introduction.

  • The second important concern I have is about the Bertrand game vs. the PG game. This is also the reason why I have chosen the PG game as an example before. While it is meaningful to talk about selection when we are talking about a stable trait, such as personality or social preferences, maybe there is not such a straight explanation for Bertrand games. The author acknowledges this in the passage below.

This difference can probably be attributed to the fact that the act of making a promise plays different roles in the public good game and the Bertrand pricing game. In the public good game, a willingness to agree to contribute more can be part of the expression of a cooperative personality. In the Bertrand pricing game this link between willingness to form an agreement and general behaviour is less straight forward. For a cooperative person it is difficult to express their cooperativeness in the regular version of the game (picking a higher number is not necessarily cooperative) and even strategic self-interested profit maximizers might have a reason to vote for a price agreement (it might increase their profits if other players adhere to it and they manage to undercut them).

After reading this paragraph I remain uncertain on what can we learn from this paper and what is the reason tha author chose a Bertrand game to investigate the research question. This is the reason why I had asked in the previous revision round to run the same experiment using a PG game. This would reveal whether the selection effect (social pref + promise keeping) is really at play. It would help if previous literature has shown that social preferences also play a role in Bertrand games. In that case, the author could maybe avoid running new experiments and find an argument to justify the choice of a Bertrand game.
